# Effect of Laser Scanning Speed on the Microstructure and Mechanical Properties of Laser-Powder-Bed-Fused K418 Nickel-Based Alloy

**DOI:** 10.3390/ma15093045

**Published:** 2022-04-22

**Authors:** Zhen Chen, Yongxin Lu, Fan Luo, Shuzhe Zhang, Pei Wei, Sen Yao, Yongxin Wang

**Affiliations:** 1State Key Laboratory of Manufacturing System Engineering, Xi’an Jiaotong University, Xi’an 710049, China; chenzhen2025@xjtu.edu.cn (Z.C.); zsz1989@stu.xjtu.edu.cn (S.Z.); weipei0529@163.com (P.W.); ysabyh@163.com (S.Y.); 2School of Material Science and Engineering, Xi’an Shiyou University, Xi’an 710065, China; lg030609@163.com; 3School of Materials Science and Engineering, Northwestern Polytechnical University, Xi’an 710072, China

**Keywords:** laser powder bed fusion, K418 alloy, orientation, phase, microhardness

## Abstract

Laser powder bed fusion (LPBF) is a powder-bed-based metal additive manufacturing process with multiple influencing parameters as well as multi-physics interaction. The laser scanning speed, which is one of the essential process parameters of the LPBF process, determines the microstructure and properties of the components by adjusting the instantaneous energy input of the molten pool. This work presents a comprehensive investigation of the effects of the laser scanning speed on the densification behavior, phase evolution, microstructure development, microhardness, and tensile properties of K418 alloy prepared by laser powder bed fusion. When the scanning speed is 800 mm/s, the microstructure of the material is dominated by cellular dendrite crystals, with coarse grains and some cracks in the melting tracks. When the scanning speed is increased to 1200 mm/s, a portion of the material undergoes a cellular dendrite–columnar crystal transition, the preferred orientation of the grains is primarily (001), and internal defects are significantly reduced. When the scanning speed is further increased to 1600 mm/s, columnar crystals become the main constituent grains, and the content of high-angle grain boundaries (HAGBs) within the microstructure increases, refining the grain size. However, the scanning speed is too fast, resulting in defects such as unmelted powder, and lowering the relative density. The experimental results show that by optimizing the laser scanning speed, the microhardness of the LPBF-ed K418 parts can be improved to 362.89 ± 5.01 HV, the tensile strength can be elevated to 1244.35 ± 99.12 MPa, and the elongation can be enhanced to 12.53 ± 1.79%. These findings could help determine the best scanning speed for producing K418 components with satisfactory microstructure and tensile properties via LPBF. In addition, since the LPBF process is largely not constrained and limited by the complexity of the geometric shape of the part, it is expected to manufacture sophisticated and complex structures with hollow, porous, mesh, thin-walled, special-shaped inner flow channels and other structures through the topology optimization design. However, due to the relatively narrow LPBF process window, this study will benefit from LPBF in producing a lightweight, complex, and low-cost K418 product, greatly improving its performance, and promoting the use of LPBF technology in the preparation of nickel-based superalloys.

## 1. Introduction

Due to its considerable creep strength, thermal and cold fatigue resistance, and oxidation resistance, K418 alloy is commonly used to produce hot precision parts operating below 900 °C, such as turbine working blades, guide vanes, guides, and whole-cast turbines [1,2,3]. Extensive research has been conducted to date on the influence of casting [4], forging [5], and injection molding [6] on the microstructure and mechanical properties of the K418 alloy. The literature has primarily concentrated on K418 alloys processed conventionally. With the increasing demand for precision components with complex structures in manufacturing, novel process techniques including laser powder bed fusion (LPBF) become worthwhile to investigate [7].

LPBF technology is a major technological approach in metallic additive manufacturing, distinguished by simple processing, the absence of tooling, high forming accuracy, short fabrication cycle, etc. It can efficiently produce parts with complex structures, and has found widespread application in the aerospace, medical, energy, and other industrial fields [8,9,10]. The metal melts and solidifies rapidly when LPBF-molding metal parts under the scanning of a high-energy laser beam, and the transition of the powder from solid to liquid and liquid to solid is extremely fast and difficult to control—especially the cooling rate of the melt pool, which can reach 10^8^ K/s [11,12,13]. Therefore, the uniqueness of the process of LPBF technology makes the durability of the material far superior to that of conventional materials. For example, Li et al. [14] studied the influence of laser scanning speed on the microstructure, phase evolution, and nanohardness of LPBF-ed Ti-45Al-2Cr-5Nb. As the scanning speed increased slightly from 500 to 800 mm/s, the nanohardness of LPBF-ed TiAl alloys was much higher than that of cast parts.

However, the parameters of the LPBF process have a complicated pattern of effects on the qualities of the formed parts, and many defects can occur with improper forming processes [15,16,17,18]. Jiang et al. [15] used LPBF to evaluate the effect of energy density on the microstructural evolution and properties of new stainless steels. The findings revealed that high scanning speeds not only caused the appearance of unmelted powder on the surface of the samples, but also increased their porosity. As a result, choosing appropriate and suitable process parameters can improve the mechanical properties. Wang et al. [19] observed the effects of scanning speed on the melt pool morphology, grain changes, and tensile properties of IN718 alloy produced by LPBF. The findings revealed that as the scanning speed increased, the melt pool’s depth-to-width ratio increased, the microstructure transformed from cellular to columnar crystals, and the tensile strength and elongation potentially exceeded 1014 ± 19 MPa and 19.04 ± 1.12%, respectively, at a laser scanning speed of 1300 mm/s.

As can be seen from the preceding literature, the scanning speed, as a critical process parameter in LPBF technology, has a significant impact on the microstructure and mechanical properties of LPBF-ed materials. In this work, K418 alloy was prepared via the LPBF technique, and the effects of scanning speed on the densification, phase composition, microstructural evolution, and mechanical behavior of LPBF specimens were thoroughly discussed. The purpose of this research is to provide more references for the additive manufacturing of nickel-based alloys.

## 2. Experimental Procedure

The K418 powder used in this work was purchased from Beijing AVIC Mighty Company. The chemical composition of the K418 powder was as follows (wt.%): 12.93 Cr, 5.96 Al, 4.26 Mo, 2.1 Nb, 0.82 Ti, 0.14 Fe, 0.13 C, and the remainder Ni. The powder had an essentially spherical morphology, with a size range of 15 to 53 µm. A stainless steel substrate with good wettability was chosen as the substrate material. The rated power of the single-mode CW fiber laser was 200 W, the laser spot diameter was approximately 60 µm, and the maximum forming size was 120 mm × 120 mm × 150 mm. Argon was employed as a protective gas to prevent the raw material from oxidizing during LPBF. This research focuses on a single variable—scanning speed; the LPBF setup parameters were determined after several trials to be laser power of 200 W, laser scanning speed of 800~1400 mm/s, layer thickness of 0.03 mm, and hatch spacing of 0.07 mm, as shown in Table 1. The scanning strategy used was a rotational scanning strategy, implying a rotation angle of 67° between the nth and n + 1st layers.

After fabrication, all LPBF-ed K418 samples were ground and polished, and the relative density of the samples was first determined using an automatic densitometer founded on Archimedes’ principle, and observed for surface defects using an optical microscope (Axio Vert. A1, Jena, Germany). The phase composition of the samples was characterized using an X-ray diffractometer (XRD-6000, Shimadzu Instrument, Kyoto, Japan) with a 20–100° diffraction angle and a scan rate of 10°/min. Following that, the samples were etched for 1 min in aqua regia (HCI:HNO_3_) = 3:1 using the standard metallographic procedure, and the microstructure of the specimens was observed using a scanning electron microscope. The grain orientation, grain boundary distribution, and grain size of the selected typical samples were analyzed using an electron backscatter diffractometer. Finally, the microhardness of the specimens’ surfaces was characterized using a microhardness tester (HXD-1000TMC, Shanghai, China), with a load of 200 g and a holding time of 10 s. Measurements were made on the surface of each specimen at 0.05 mm intervals, with 15 measurement points, and their average values were calculated. The tensile properties of the LPBF samples were tested using a tensile testing machine at a stretching rate of 0.5 mm/min, in accordance with ASTM D638 GB/T 228-2010. To ensure data accuracy, three samples were tested for each parameter. After the tensile experiments, scanning electron microscopy was used to characterize the tensile fracture morphology.

## 3. Results and Discussion

### 3.1. Densification Behavior

Figure 1 shows the densities of the LPBF samples of K418 alloy at various scanning speeds, as well as the corresponding metallographic images. It can be seen that the relative density of the LPBF-ed K418 specimens increased slowly at first, and then rapidly as the laser scanning speed increased. When the laser scanning speed was reduced to 800–1000 mm/s, the densities were approximately 97.6%, and the corresponding optical photographs show that there were tiny pores and cracks on the surfaces of the samples. At a laser scanning speed of 1200 mm/s, the densities reached 97.76%, and the optical photographs revealed the best surface morphology of the samples. As the scanning speed increased to 1400–1600 mm/s, the surface of the samples became more porous and less well-formed, and the densities decreased rapidly. As a result, when the scanning speed was too slow, the laser beam stayed in the molten pool for an extended period of time, causing the energy input in the molten pool to be too high, and resulting in turbulence and instability of the melt in the molten pool which, in turn, resulted in porosity and balling, along with other defects, thereby reducing the relative density [20]. However, when the scanning speed was too fast, the molten pool formation and cooling time were very short, the Marangoni convection strength in the molten pool was weak, and the melt was prone to Plateau–Rayleigh instability due to surface tension and capillary force. Discontinuous melt tracks with more unmelted or semi-melted metal particles result in holes and cracks on the surface, reducing the material’s densification.

### 3.2. Phase Behavior

Figure 2 shows the XRD analysis of K418 samples prepared at different scanning speeds. The XRD results show that as the scanning speed increased, the changes in the γ phase and γ′/γ″ phase both became available in various grain orientations. As the key strengthening phase is the γ′/γ″ phase in Ni-based alloys, these are the strongest peak and the weakest peak, respectively. In addition, the sample grain size is proportional to the XRD diffraction peak’s full width at half-maximum (*FWHM*), based on Scherrer and Wilson’s equation formulae [21]:(1)FWHM=0.9μ/Dcosθ
where *FWHM* is the diffraction peak, *μ* is the X-ray radiation wavelength, *D* denotes the mean grain size, and θ is the scanning angle. Therefore, on the strongest peak, the half-width of the γ′/γ″ phase diffraction peak at different scanning speeds can be calculated using X’Pert HighScore Plus software (Royal Dutch Philips, Amsterdam, The Netherlands), as illustrated in Table 2. When the laser scanning speed was 1200 mm/s, the half-width of γ′/γ″ phase diffraction peak was the largest, the average grain size of the material was reduced, and the properties of the material were enhanced [22,23]. In addition, no carbide peaks were detected in the XRD results, indicating a low level of internal carbide generation.

### 3.3. Microstructure Evolution

Figure 3 depicts the metallographic microstructure of K418 samples in the corrosion state as a function of scanning speeds. As the scanning speed increased, the morphology and arrangement of the molten pool changed. The laser input energy density was high at low scanning speeds, causing the temperature of the molten pool to rise too quickly, and the Marangoni convection intensity inside the molten pool to be too strong to produce turbulent flow, resulting in large fluctuations in the morphology of the forming surface and the laser energy absorbed by the K418 powder [24,25]. When the scanning speed was 1200 mm/s, the thermal flow of the molten pool was stable, the quality of the melt track quality improved, and cracks were reduced. However, as the scanning speed increased, there were defects such as unmelted powder and porosity on the surface of the melt pool, as well as the melt pool being short and irregular due to the faster laser shift speed, which cannot completely melt the powder.

Figure 4 illustrates SEM images of the K418 samples at various scanning speeds. At low scanning speeds, the specimen had a cellular dendritic microstructure inside the specimen, due to the increased residence time of the laser beam in a specific area, the restricted rate of energy loss in the melt pool, and the prolonged cooling time, promoting grain nucleation and growth. When the scanning speed was increased to 1200 mm/s, some of the microstructures in the specimen were converted to columnar crystals, which noticeably refined and grew continuously. However, increasing the scanning speed shortens the action time of the laser beam on the surface of the molten pool, accelerates heat conduction in the molten pool, and reduces the solidification time of the molten pool, providing suitable solidification conditions for the formation of columnar crystals. As a result, columnar crystals can be seen to preferentially grow along the deposition direction [26].

Based on metallographic, scanning, and XRD analysis, it is clear that the microstructure transitions from cellular dendrites to columnar crystals, that the columnar crystals are obviously refined, and that the γ′/γ″-reinforced phase is formed at the boundary of the columnar crystals as the scanning speed increases. According to solidification theory, the cooling rate of the melt pool is positively correlated with the laser scanning speed during LPBF, and the phase transition of the microstructure during LPBF is strongly related to the cooling rate during solidification [24]. As a result, as the scanning speed increases from 800 mm/s to 1600 mm/s, the cooling rate tends to increase. Microstructure can be determined according to Equation (2) [27]:(2)$$\lambda_{1}=A\left(G^{-0.55}\cdot V^{-0.28}\right)$$
where *λ*_1_ is the primary dendrite spacing, *A* is a constant, *G* represents the temperature gradient, and *V* is the cooling rate. The primary dendrite spacing is inversely proportional to the cooling rate. Therefore, as the scanning speed increases, the interaction time between the laser beam and the powder decreases, the temperature of the center of the molten pool decreases, and the cooling rate increases, resulting in a decrease in primary dendrite spacing. Furthermore, when the scanning speed is too slow, the cooling rate of the molten pool is slow, there are more liquid phases in the molten pool, and the unmelted particles surrounding the molten pool are quickly dragged into the molten pool due to strong Marangoni convection and high surface tension, resulting in interlayer defects. Thus, residual stress is generated, resulting in the formation of interdendritic thermal cracks. When the scanning speed is too fast, the laser moves too quickly, the temperature of the molten pool is low, the cooling rate is high, and the surrounding powder particles in the molten pool are too late to melt completely, resulting in the occurrence of defects such as non-fusion and pores, as well as a large amount of accumulation in these defects. Furthermore, as the laser energy density increases, the LPBF-ed microstructure preferentially grows ultrafine columnar crystals rather than secondary dendrites, and the microstructure has a significant impact on the mechanical properties [28]. The XRD spectra in Figure 2 clearly show that the scanning speed influences the intensity of the diffraction peaks, implying that the grain orientation has shifted. As a result, the EBSD results at various scanning speeds differ significantly. It can be clearly seen from Figure 5 that the microstructure at 800 mm/s is composed of coarser cellular dendrite crystals with a balanced microstructure color distribution. Figure 6 depicts the grain orientation of a longitudinal section of K418 alloy at various scanning speeds. The weave intensity of the material is 2.93, as shown by its corresponding antipodal map (Figure 6a), indicating weak anisotropy. The red color inside the microstructure increases at a scanning speed of 1200 mm/s, implying that the crystals grow significantly in the first (001) orientation meritocracy. Furthermore, the inverse polarographic projection (Figure 6b) relationship reveals that the microstructure has a high weave strength, with an intensity factor of 6.72. The number of fine crystals within the microstructure increases as the scanning speed increases to 1600 mm/s, but the red area decreases slightly, indicating that the (001) directional selective orientation is weakened. The weakened orientation is also demonstrated by the weaving intensity in the corresponding anti-polar diagram in Figure 6c.

The difference in grain boundary orientation is also an important indicator for the microstructural characterization of the material. Figure 7 depicts the distribution of grain boundaries in K418 alloy at various laser scanning speeds. The microstructure of the specimen is dominated by high-angle grain boundaries (HAGBs). The preparation of samples layer by layer and row by row is well known to be an important feature of the LPBF technique. As the laser scans the next layer, the previously solidified melt pool is partially remelted, similar to the annealing heat treatment process, which inevitably results in recrystallization and, thus, a high percentage of HAGBs [29]. However, the migration and integration rate of subgrain boundaries decreases as the laser scanning speed increases, resulting in a reduction in HAGBs. As a result, as shown in Figure 7e, the grain size decreases as the scanning speed increases.

Figure 8 depicts the Schmidt factor and distribution statistics for the K418 alloy at various scanning speeds. According to Schmidt’s law, a smaller Schmidt factor results in a higher yield strength [30]. Figure 8 shows that the Schmidt factors are approximate at different scanning speeds and, thus, their yield strengths do not differ significantly. As the laser scanning speed increases from 800 mm/s to 1600 mm/s, the migration and integration rates of the subgrain boundaries decrease, causing the HAGB content to increase from 40.4% to 57.2% and the grain size to increase from 0.99 μm to 1.29 μm, as shown in Table 3.

### 3.4. Microhardness

The microhardness of LPBF-ed K418 samples at various scanning speeds is depicted in Figure 9. When the scanning speed is low, the energy heat input is high, resulting in coarse internal grains of the microstructure and low hardness. As the scanning speed increases, the grains are refined, the melt pool morphology tends to be uniform, and the extremely fast cooling rate causes the matrix grains to grow too late, resulting in fine crystal reinforcement and high hardness. The linear relationship between Vickers hardness and grain size is defined by Hall and Petch as follows [31]:(3)$$H_{v}=H_{0}+K_{H}D^{-\frac{1}{2}}$$
where *H*_0_ and *K_H_* are the corresponding constants. The equation clearly shows that hardness is positively related to grain size. Finer grain size results in higher hardness values. When the scanning speed is too fast, the laser energy input is relatively low, resulting in discontinuous melting tracks or unmelted powder, internal defects, and a reduction in the microhardness of the formed part.

### 3.5. Tensile Property

Due to the extremely high temperature gradient and cooling rate (~10^8^ K/s) of the LPBF processing, the grain size after solidification is at the micrometer scale. Simultaneously, the material tends to generate metastable phases that deviate from the equilibrium state due to the repeated thermal cycling process of rapid heating and rapid cooling. The microstructural evolution is largely determined by the temperature gradient in the molten pool, whereas the temperature gradient is influenced by the scanning speed, which affects the microstructural evolution characteristics by manipulating the local solidification conditions of the molten pool. The diffusion process is frequently constrained by the narrow solidification window, second-phase precipitation or component segregation is rare, and fine columnar crystals are obtained, inhibiting the growth of secondary dendrite arms and ensuring excellent mechanical properties of LPBF-ed K418 parts. On the other hand, since nickel has a face-centered cubic (FCC) crystal structure, the maximum heat flow and temperature gradient are generally consistent with the deposition direction during LPBF processing—that is, from the substrate along the direction of the build-up layer height increase. Therefore, the microstructure of LPBF-ed K418 alloy tends to exhibit <100> cubic-textured columnar grains, which influence the final mechanical properties. Figure 10 depicts the tensile properties of K418 samples at various scanning speeds, demonstrating that the tensile strength and elongation of the samples tend to increase and then decrease as the scanning speed increases, whereas the yield strength shows only a slight change. The tensile properties were the highest at 1200 mm/s, with the tensile strength reaching 1244.35 ± 99.12 MPa, the yield strength reaching 863.89 ± 132.71 MPa, and the elongation of the material reaching a maximum of 12.53 ± 1.79%. When the scanning speed is less than 1200 mm/s, the powder absorption time is longer, the melt pool cooling time is slower, and the subsequent molten powder is prone to sputtering, resulting in poor melt track bonding. When the scanning speed exceeds 1200 mm/s, the scanning track becomes discontinuous, the metal powder is unable to be completely melted, and defects such as pores and unmelted particles appear. When loaded, the defects act as a source of crack propagation, resulting in early failure and reduced ductility.

Figure 11 depicts the SEM morphology of a tensile fracture at various scanning speeds. When the scanning speed is low, a large number of holes as well as cracks appear in the fracture morphology, resulting in lower strength of the material. When the scanning speed is increased to 1200 mm/s, the defects are obviously reduced, the fractures are distributed with a certain number of tough nests, and the typical feature-tearing ribs can also be clearly observed, as shown in Figure 11c. Then, as the scanning speed increases, a certain number of unmelted particles and pores appear at the fracture. Figure 12 depicts a schematic diagram of the defect distribution in the tensile fracture cross-section corresponding to different laser powers. According to Figure 12 and the macroscopic morphology of the tensile fracture, it can be concluded that the material’s tensile properties are related to internal defects such as unmelted particles, pores, and cracks. At a low scanning speed (800–1000 mm/s), defects such as pores and cracks exist in the internal microstructure and grain boundaries, fracturing the specimen and reducing its tensile properties. However, due to the presence of numerous small uniform dendrites and fewer defects in the microstructure, the ideal tensile properties are obtained at a scanning speed of 1200 mm/s. Although the internal microstructure of the specimen was significantly refined when the scanning speed was increased (1400–1600 mm/s), the cooling rate was too fast due to the excessive scanning speed, and the powder melted incompletely, resulting in unmelted particles, pores and other defects, which generated stress concentration and, thus, reduced the tensile properties [32].

## 4. Conclusions

In this work, the effects of laser scanning speed on the relative density, microstructural characteristics, phase transformation, microhardness, and tensile properties of LPBF-ed K418 samples were investigated. The main findings are summarized as follows:(1)There are numerous factors affecting the LPBF processing, and the scanning speed is one of the key process parameters influencing the microstructure and properties of the LPBF-ed K418 superalloy. A satisfactory tensile strength of 1244.35 ± 99.12 MPa and elongation of 12.53 ± 1.79% were obtained through process optimization.(2)As the scanning speed increased from 800 mm/s to 1600 mm/s, the tensile strength and elongation of the material tended to first increase and then decrease, while the yield strength remained stable. The optimal comprehensive mechanical properties were obtained when the scanning speed was 1200 mm/s.(3)The microstructure of LPBF-ed K418 exhibits directional preferential growth of columnar crystals along the (001) direction, which is dominated by high-angle grain boundaries (HAGBs), and the volume fraction of HAGBs decreases with increasing scanning speed.(4)The optimized laser scanning speed can be used as a reference for the LPBF preparation of nickel-based superalloys, and it is expected that this study will promote the industrial application of nickel-based superalloys prepared via LPBF.

## Figures and Tables

**Figure 1 materials-15-03045-f001:**
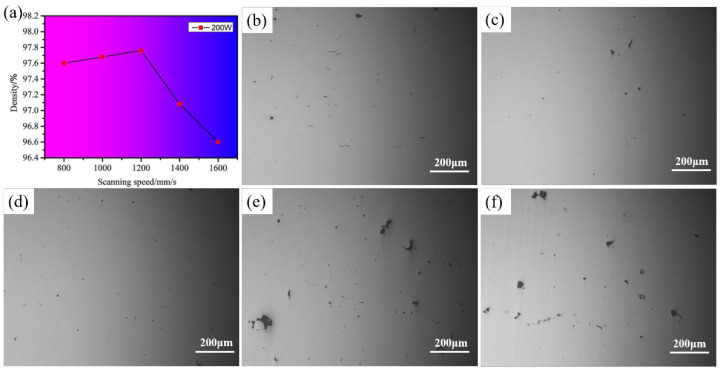
(**a**) Influence of different scanning speeds on relative density and (**b**–**f**) transverse section appearance: (**b**) 800 mm/s; (**c**) 1000 mm/s; (**d**) 1200 mm/s; (**e**) 1400 mm/s; (**f**) 1600 mm/s.

**Figure 2 materials-15-03045-f002:**
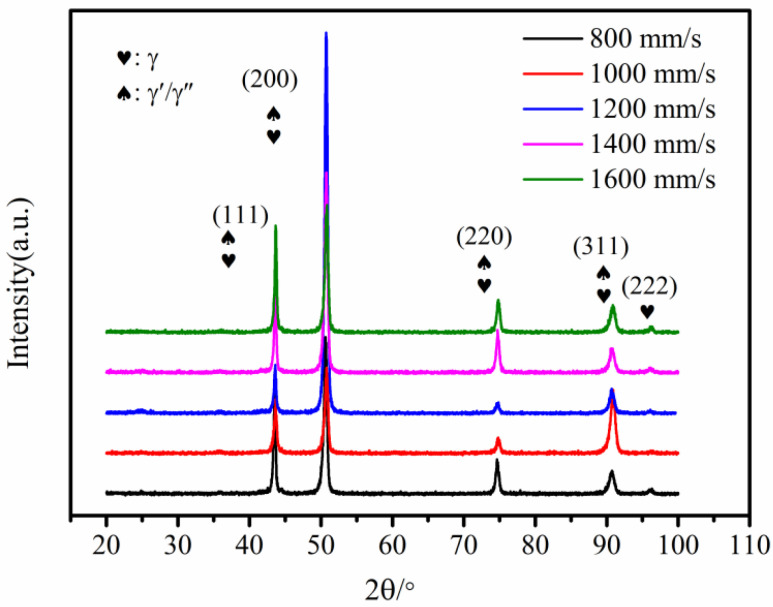
XRD analysis of K418 samples prepared at different scanning speeds.

**Figure 3 materials-15-03045-f003:**
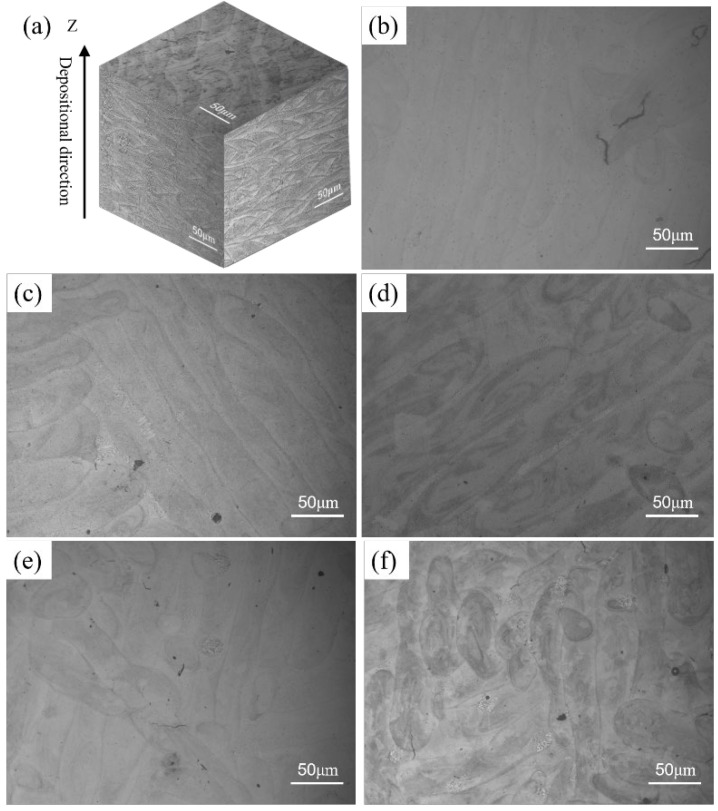
(**a**) The 3D metallographic diagram of k418, and (**b**–**f**) the transverse section metallography corresponding to different laser powers ((**b**) 800 mm/s; (**c**) 1000 mm/s; (**d**) 1200 mm/s; (**e**) 1400 mm/s; (**f**) 1600 mm/s).

**Figure 4 materials-15-03045-f004:**
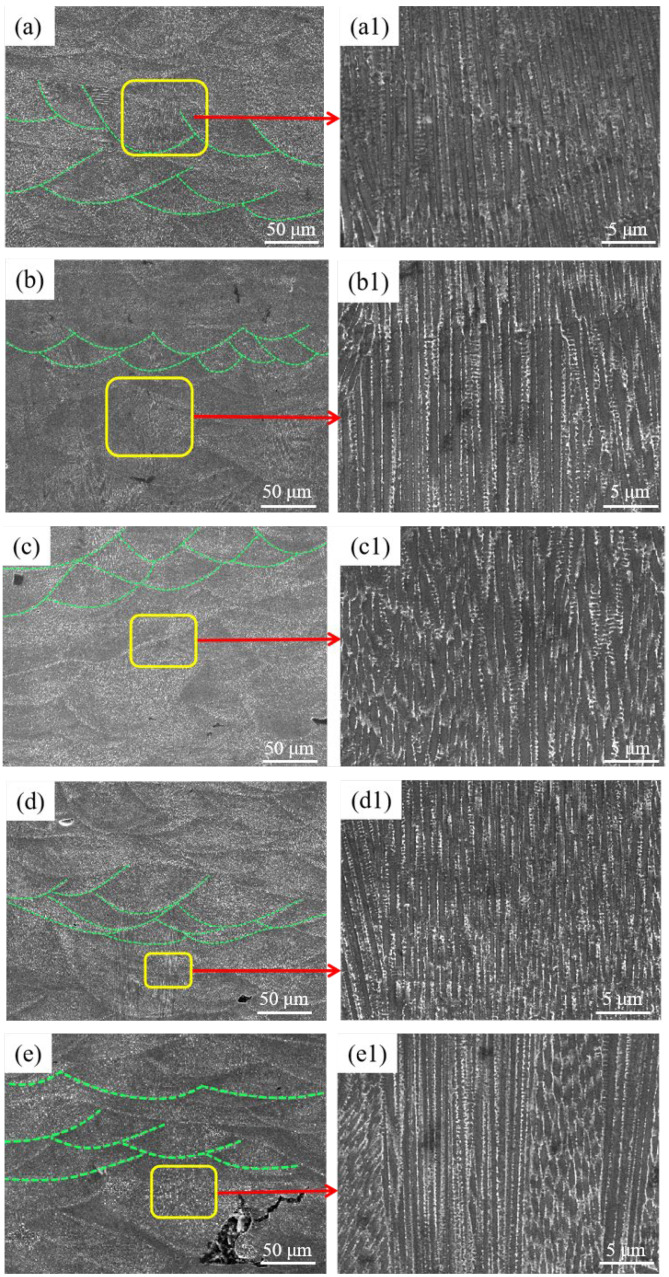
SEM image of the corresponding K418 sample under different scanning speeds: (**a**,**a1**) 800 mm/s; (**b**,**b1**) 1000 mm/s; (**c**,**c1**) 1200 mm/s; (**d**,**d1**) 1400 mm/s; (**e**,**e1**) 1600 mm/s.

**Figure 5 materials-15-03045-f005:**
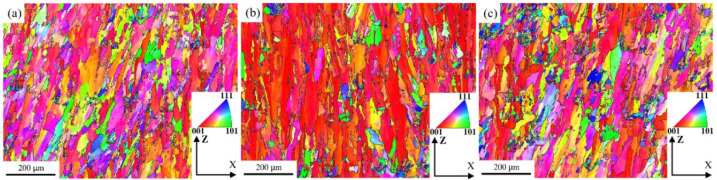
Grain orientation maps of K418 alloy at different scanning speeds: (**a**) 800 mm/s; (**b**) 1200 mm/s; (**c**) 1600 mm/s.

**Figure 6 materials-15-03045-f006:**
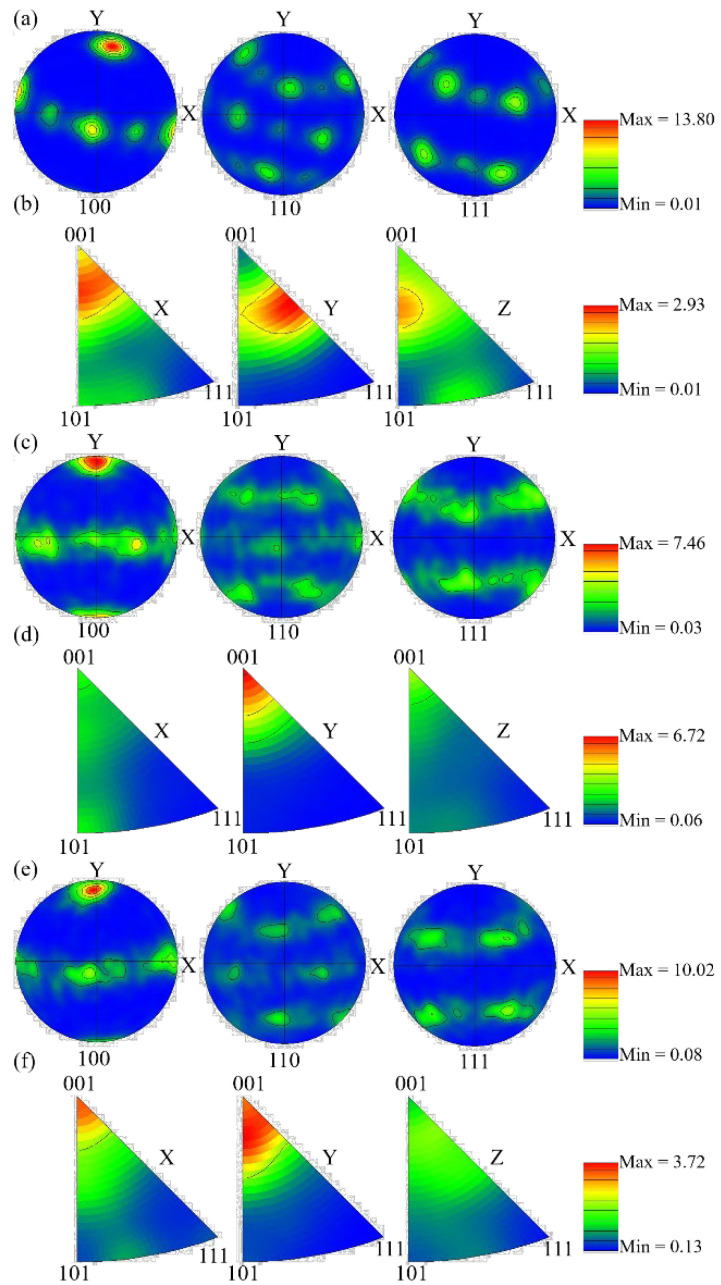
The polar diagrams ((**a**) 800 mm/s; (**c**) 1200 mm/s; (**e**) 1600 mm/s) and anti-polar diagrams ((**b**) 800 mm/s; (**d**) 1200 mm/s; (**f**) 1600 mm/s) of K418 alloy at different scanning speeds.

**Figure 7 materials-15-03045-f007:**
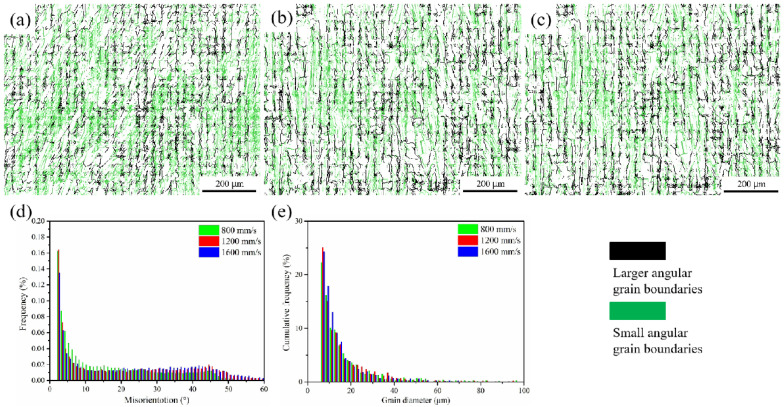
Grain boundary distribution of K418 alloy at different scanning speeds ((**a**) 800 mm/s; (**b**) 1200 mm/s; (**c**) 1600 mm/s), (**d**) grain boundary size statistics of K418 alloy at different scanning speeds, and (**e**) grain size statistics of K418 alloy at different scanning speeds.

**Figure 8 materials-15-03045-f008:**
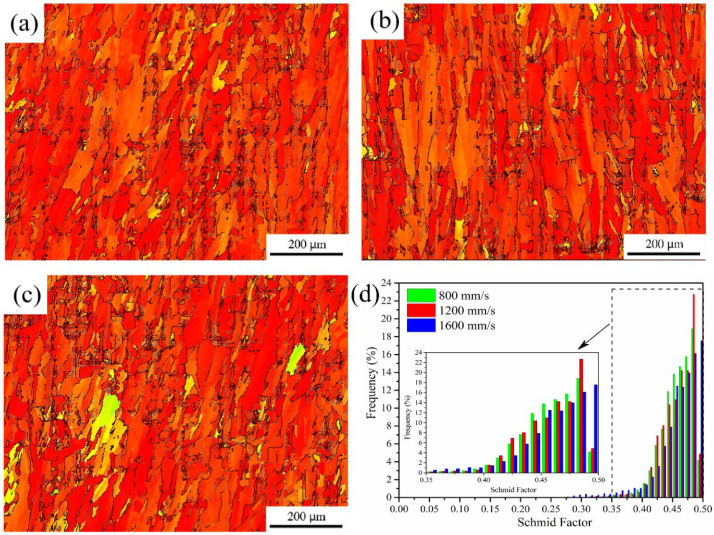
Schmidt factor distribution ((**a**) 800 mm/s; (**b**) 1200 mm/s; (**c**) 1600 mm/s) and (**d**) statistical diagram of K418 alloy at different scanning speeds.

**Figure 9 materials-15-03045-f009:**
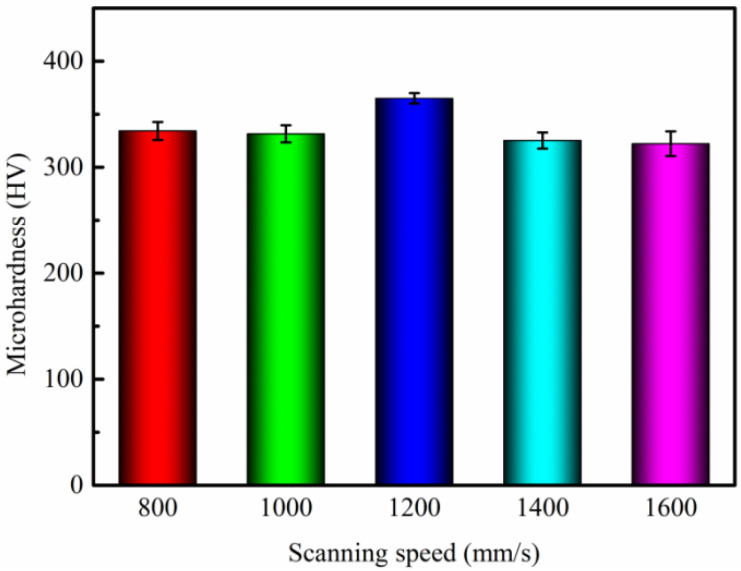
Influence of different scanning speeds on the microhardness of K418.

**Figure 10 materials-15-03045-f010:**
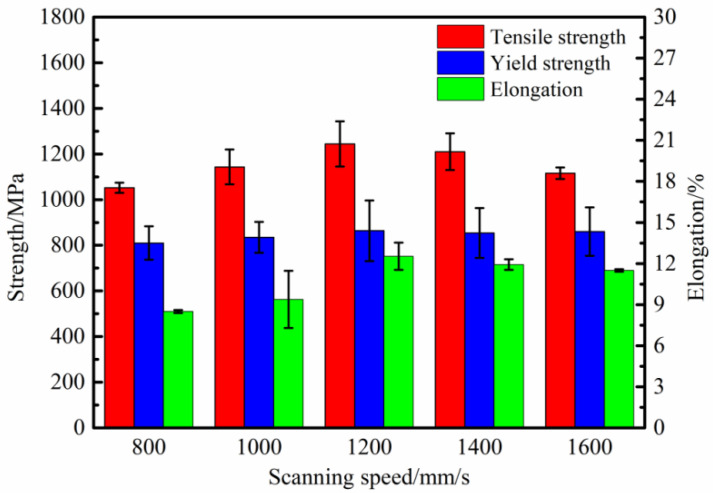
Influence of different scanning speeds on the tensile properties of K418.

**Figure 11 materials-15-03045-f011:**
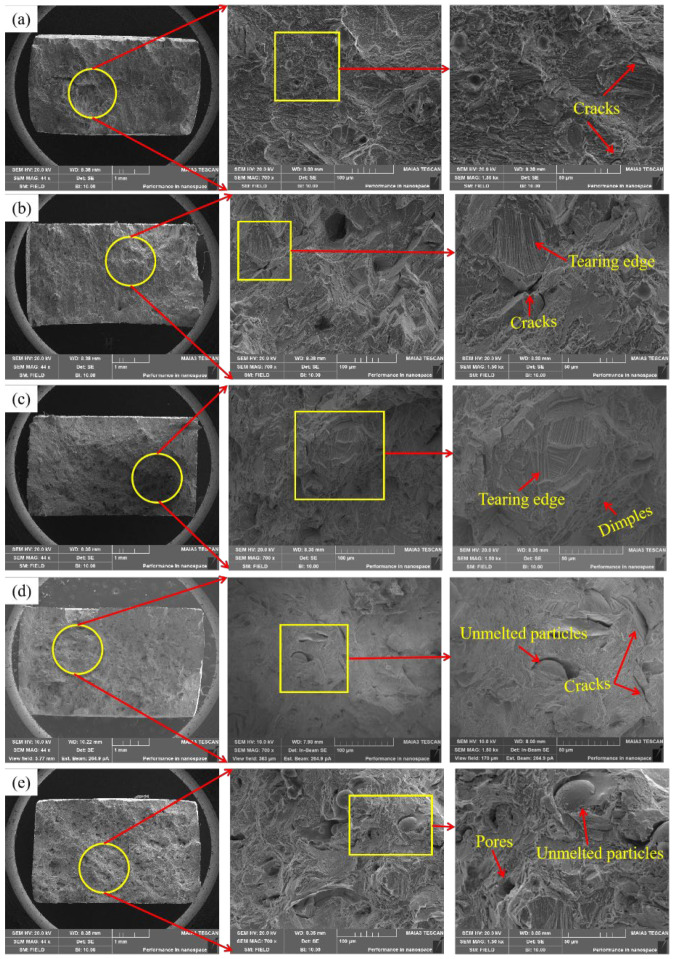
SEM morphology of tensile fractures corresponding to different laser powers ((**a**) 800 mm/s; (**b**) 1000 mm/s; (**c**) 1200 mm/s; (**d**) 1400 mm/s; (**e**) 1600 mm/s).

**Figure 12 materials-15-03045-f012:**
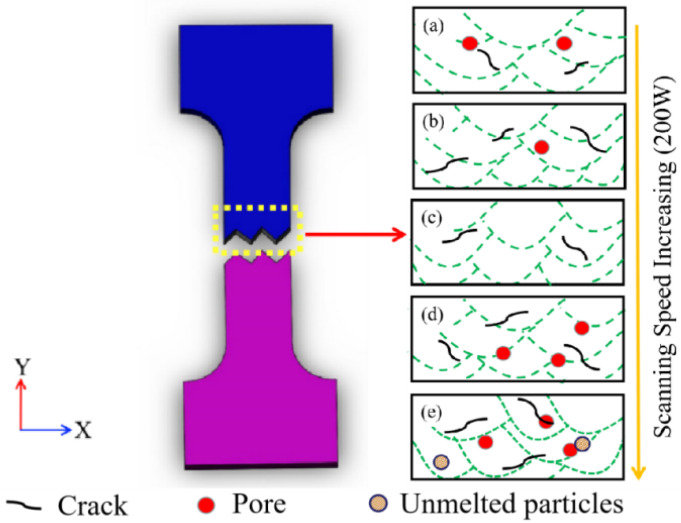
Defect distribution diagrams of tensile fractures’ transverse sections corresponding to different laser powers ((**a**) 800 mm/s; (**b**) 1000 mm/s; (**c**) 1200 mm/s; (**d**) 1400 mm/s; (**e**) 1600 mm/s).

**Table 1 materials-15-03045-t001:** Laser powder bed fusion process parameters.

Process Parameters	Laser Power (W)	Scanning Speed (mm/s)	Layer Thickness (mm)	Hatch Spacing (mm)
Value	200	800, 1000, 1200, 1400, 1600	0.03	0.07

**Table 2 materials-15-03045-t002:** *FWHM* of γ′/γ″ phase diffraction peaks at different scanning speeds.

Scanning Speeds (mm/s)	800	1000	1200	1400	1600
*FWHM*	0.1968	0.4330	0.4723	0.1181	0.1378

**Table 3 materials-15-03045-t003:** Statistics of EBSD test results under different scanning speeds.

Scanning Speeds (mm/s)	HAGBs	Average Grain Size	Substructured	Deformed
800	40.4%	0.99 μm	16.67%	75.29%
1200	52.5%	1.01 μm	13.07%	80.23%
1600	57.2%	1.29 μm	13.22%	75.97%

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
