# Peer review of "Effect of Laser Scanning Speed on the Microstructure and Mechanical Properties of Laser-Powder-Bed-Fused K418 Nickel-Based Alloy"

_materials, 2022, doi:10.3390/ma15093045_

Round 1
Reviewer 1 Report
- Since you have done enough work on process conditions, crystal structure, and mechanical properties, you should include a Discussion section to discuss the correlation between them quantitatively.
- Both the abstract and conclusion simply describe the experimental results, which is insufficient in terms of novelty. You should describe why different scanning velocities resulted in different crystal orientations and mechanical properties, and provide a more abstract and generic mechanism. Otherwise, the paper will be considered valuable only for your equipment and your material.
- The Densification behavior in 3.1 is not new in itself, as it has been observed in other nickel alloys and other metals. In your paper, you describe the mechanism of defect formation in line 119-125, but you should include existing studies instead of writing your imagination. There are many reports on the behavior of powder melting using numerical analysis and advanced observation techniques. line 153-159 173–176, 193–199 should be also improved. I am skeptical of your manuscripts which are only your imagination.
- I don't understand what you are trying to say in Figure 6. For example, what is the clear difference between (a) and (e)? Also, does this figure correspond to the actual microstructure observed in Figure 5?
- I do not know what Figure 9(a)(b)(c) shows. Please explain properly and highlight the differences between them. Also, the graphs in (d) and (e) appear to have the same distribution at 800, 1200, and 1600 mm/s.
- At what temperature and for how long should K418 recrystallization occur? Can that happen with the SLM process?
- Please quantify whether equations (2) and (3) also apply to your sample results. To apply equation (2), have you calculated the temperature gradient and cooling rate at varying scanning speeds?Have you quantitatively evaluated the microstructure to apply the Hall-Petch relationship equation? You should evaluate them.
- Figure 14 is also not a newly discovered fact. As in this experiment, many researchers have already reported the formation of spherical defects and insufficient melting at different scanning speeds, which affect tensile properties. After all, what is new?
Author Response
Dear professor,
We sincerely appreciate all of your insightful comments and criticism. Those comments are all valuable and very helpful in modifying and improving our paper, as well as having important guiding significance for our research. We studied the comments carefully and made detailed modifications to the full text to avoid grammatical errors, and the English has been polished. These modifications should have no effect on the paper's content or framework. The revised segments are highlighted in the revised manuscript. for more information, please see the attachment.
Please do not hesitate to contact us if you have any questions.
Sincerely yours
Zhen Chen
chenzhen05lg810@163.com; chenzhen2025@tjtu.edu.cn
State Key Laboratory of Manufacturing System Engineering
Xi'an Jiaotong University
Xi’an 710049, China

Reviewer 2 Report
The manuscript explains the effect of laser scanning speed on microstructure and mechanical properties of SLM-formed K418 Nickel-based alloy. The manuscript needs an extensive revision before considering for publication. The following comments should be considered in the revised version:
1- The title of the manuscript is misleading. Please Revise that. What is the scanning speed? You meant laser scanning speed. What is SLM standing for? It is ambiguous and readers cannot understand it easily.
2- There are many typos and grammatical mistakes. Please revise the whole manuscript, and the language should be checked by an English native speaker who is an expert in the field. Also, in citations, there are some errors such as: line 56 Pfj A?
3- The explanation about the mechanism of grain size change and recrystallization is not adequate. It should be explained based on the references. Also, the reason why the fraction of HAGBs changed should be clarified.
4- in the introductory part, the microstructure of the nickel-based alloy should be explained and introduced in detail. Other references should be added to the paper. https://doi.org/10.1016/j.vacuum.2019.108890
https://doi.org/10.5006/1.35851105- The conclusions are not representative of the results. Please extend it, and complete the conclusions.
6- The relationship between texture and scanning speed is not explained satisfactorily. The mechanism should be explained in detail.
Author Response

(The authors gave the same response as above.)

Round 2
Reviewer 1 Report
-
As in my previous comments 1 and 2, it turns out that K418 in your study was not a model material for investigating the phenomenon of nickel-based alloys in general, but simply the subject of your interest. It would have been nice if the abstract and conclusions were appealing to readers who are not interested in K418.
- As in my previous comment 3, please add some journal papers to support your thought for lines 120–128, 153–164, 175–180, and 195–211.
- As in my previous comment 4, I still do not understand what Figure 5 shows. All microstructures in Figs. 4 (a1) to (e1) appear to correspond to Fig. 5 (e). In other words, your schematic diagram in Fig. 5 does not properly represent the microstructure and defects actually observed.
- As in my previous comment 5, Figures 9 (and 10) do not appear to show a clear difference in scanning velocity, and are suspected to be within the margin of error. 1000 mm/s and 1400 mm/s samples should also be analyzed to confirm the trend. The bar graphs are very confusing. Please improve them.
- As for my previous comment 6, please show us some journal papers to support your idea (dynamic recrystallization may occur during this process, resulting in non-equilibrium microstructures). In addition, please carefully investigate the previous studies about K418 for welding, casting, forging, and so on. Again, at what temperature and for how long should K418 recrystallization occur? Can that happen with the SLM process?
- As for comment 7, please refer to some journal paper to investigate the temperature gradient and the cooling rate with different scanning speeds, and substitute the values into Equation 2 to verify your primary dendrite spacing. Please tell us what size the observed and derived spacings are. You have added Table 3 and the grain sizes increased with the increase of the scanning speeds. However, the hardness was highest at 1200 mm/s. In your manuscript, you think that this may be due to discontinuous melting tracks, unmelted powder, internal defects . Did you consider the indenter size of vickers hardness test? The indenter was pushed in the defects? I guess you avoided testing at locations with such defects.
Author Response
Dear professor,
We are very grateful to you for all your insightful comments and criticism. Those comments are all valuable and very helpful to the modification and improvement of our paper, and also have important guiding significance for our research. We have carefully studied the comments and corrected them, hoping to get approval. The revised portions are highlighted in the revised manuscript. The main corrections and responses are as follows:
- As in my previous comments 1 and 2, it turns out that K418 in your study was not a model material for investigating the phenomenon of nickel-based alloys in general, but simply the subject of your interest. It would have been nice if the abstract and conclusions were appealing to readers who are not interested in K418.
Response:
According to your professional and insightful guidance, we have reorganized the abstract and conclusion to better reflect the innovation and contribution of the article, as follows:
Abstract: Selective laser melting is a powder bed-based metal additive manufacturing process with multiple influencing parameters as well as multi-physics interaction. The laser scanning speed, which is one of the essential process parameters of the SLM process, determines the microstructure and properties of the components by adjusting the instantaneous energy input of the molten pool. This work presents a comprehensive investigation of the effect of the laser scanning speed on the densification behavior, phase evolution, microstructure development, microhardness, and tensile properties of K418 alloy prepared by selective laser melting (SLM). When the scanning speed is 800 mm/s, the microstructure of the material is dominated by cellular dendrites crystals, with coarse grains and some cracks in the melting tracks. When the scanning speed is increased to 1200 mm/s, a portion of the material undergoes a cellular dendrites-columnar crystal transition, the preferred orientation of the grains is primarily (001), and internal defects are significantly reduced. When the scanning speed is further increased to 1600 mm/s, columnar crystals become the main constituent grains, and the content of high-angle grain boundaries (HAGBs) within the microstructure increases, refining the grain size. However, the scanning speed is too fast, resulting in defects such as un-melted powder and lowering the density. The experimental results show that by optimizing the laser scanning speed, the microhardness of the SLMed K418 parts can be improved to 362.89 ± 5.01 HV, and tensile strength can be elevated to 1244.35 ± 99.12 MPa, and elongation can be enhanced to 12.53 ± 1.79%. These findings could help determine the best scanning speed for producing K418 components with satisfactory microstructure, and tensile properties by SLM. In addition, since the SLM process is almost not constrained and limited by the complexity of the geometric shape of the part, it is expected to manufacture sophisticated and complex structures with hollow, porous, mesh, thin-walled, special-shaped inner flow channels and other structures through the topology optimization design. However, because the SLM process window is relatively narrow, this study will benefit from SLM in realizing the lightweight, complex, and low-cost K418 product, greatly improving its performance and promoting SLM technology in the preparation of nickel-based superalloys.
Conclusions: In this work, the effect of laser scanning speed on the density, microstructural characteristics, phase transformation, microhardness, and tensile properties of SLM-produced K418 samples was investigated. The main findings are summarized as follows.
(1) There are numerous factors affecting the SLMprocessing, and the scanning speed is one of the key process parameters influencing the microstructure and properties of the SLM forming K418 superalloy, a satisfactory tensile strength of 1244.35 ± 99.12MPa and elongation of 12.53 ± 1.79% were obtained through process optimization.
(2) As the scanning speed increased from 800 mm/s to 1600 mm/s, the tensile strength and elongation of the material tended to increase firstly and then decrease, while the yield strength remained stable, the optimum comprehensive mechanical properties were obtained when the scanning speed was 1200 mm/s.
(3) The microstructure of SLM-formed K418 exhibits directional preferential growth of columnar crystals along the (001) direction, which is dominated by high-angle grain boundaries (HAGBs), and the volume fraction of HAGBs decreases with increasing scanning speed.
(4) The optimized laser scanning speed can be used as a reference for the SLM preparation of nickel-base superalloys, and it is expected that this study will promote the industrial application of nickel-base superalloys prepared by SLM.
- As in my previous comment 3, please add some journal papers to support your thought for lines 120–128, 153–164, 175–180, and 195–211.
Response:
According to your professional guidance, we have added some journal papers for lines 120–128, 153–164, 175–180, and 195–211.
- As in my previous comment 4, I still do not understand what Figure 5 shows. All microstructures in Figs. 4 (a1) to (e1) appear to correspond to Fig. 5 (e). In other words, your schematic diagram in Fig. 5 does not properly represent the microstructure and defects actually observed.
Response:
According to your professional guidance, we have deleted figure 5.
- As in my previous comment 5, Figures 9 (and 10) do not appear to show a clear difference in scanning velocity, and are suspected to be within the margin of error. 1000 mm/s and 1400 mm/s samples should also be analyzed to confirm the trend. The bar graphs are very confusing. Please improve them.
Response:
According to your professional guidance, we have deleted figure 9 and modified figure 10.
Figure 10. Schmid factor distribution ((a)800 mm/s; (b)1200 mm/s and (c)1600 mm/s) and statistics diagram (d) of K418 alloy with different scanning speeds
- As for my previous comment 6, please show us some journal papers to support your idea (dynamic recrystallization may occur during this process, resulting in non-equilibrium microstructures). In addition, please carefully investigate the previous studies about K418 for welding, casting, forging, and so on. Again, at what temperature and for how long should K418 recrystallization occur? Can that happen with the SLM process?
Response:
According to your professional guidance, we have deleted this part.
- As for comment 7, please refer to some journal paper to investigate the temperature gradient and the cooling rate with different scanning speeds, and substitute the values into Equation 2 to verify your primary dendrite spacing. Please tell us what size the observed and derived spacings are. You have added Table 3 and the grain sizes increased with the increase of the scanning speeds. However, the hardness was highest at 1200 mm/s. In your manuscript, you think that this may be due to discontinuous melting tracks, unmelted powder, internal defects . Did you consider the indenter size of vickers hardness test? The indenter was pushed in the defects? I guess you avoided testing at locations with such defects.
Response:
According to your professional guidance, we checked the paper and experimental data again, and hardness measurements were made on the surface of each specimen at 0.05 mm intervals with 15 measurement points, and the final hardness value is the average of 15 measurement points.
We tried our best to improve the manuscript and made some changes to the revised manuscript. We try to avoid grammatical errors, and the English has been polished. These modifications should have no impact on the content and framework of the paper.
If you have any questions, please do not hesitate to contact us.
Sincerely yours
Zhen Chen
chenzhen05lg810@163.com; chenzhen2025@tjtu.edu.cn
State Key Laboratory of Manufacturing System Engineering
Xi'an Jiaotong University
Xi’an 710049, China

Reviewer 2 Report
The revised version is much better than the previous one. The manuscript still needs a language and style revision.
Author Response
Dear professor,
We are very grateful to you for all your insightful comments and criticism. Those comments are all valuable and very helpful to the modification and improvement of our paper, and also have important guiding significance for our research. We have carefully studied the comments and corrected them, hoping to get approval. The revised portions are highlighted in the revised manuscript.
We tried our best to improve the manuscript and made some changes to the revised manuscript. We try to avoid grammatical errors, and the English has been polished. These modifications should have no impact on the content and framework of the paper.
If you have any questions, please do not hesitate to contact us.
Sincerely yours
Zhen Chen
chenzhen05lg810@163.com; chenzhen2025@tjtu.edu.cn
State Key Laboratory of Manufacturing System Engineering
Xi'an Jiaotong University
Xi’an 710049, China
